# Non-Targeted LC-MS/MS Assay for Screening Over 100 Lipid Mediators from ARA, EPA, and DHA in Biological Samples Based on Mass Spectral Fragmentations

**DOI:** 10.3390/molecules24122276

**Published:** 2019-06-19

**Authors:** Gabriel Dasilva, Silvia Muñoz, Salomé Lois, Isabel Medina

**Affiliations:** 1Instituto de Investigaciones Marinas, Consejo Superior de Investigaciones Científicas (IIM-CSIC), E-36208 Vigo, Galicia, Spain; silviam@iim.csic.es (S.M.); salomel@iim.csic.es (S.L.); medina@iim.csic.es (I.M.); 2Department of Analytical Chemistry, Nutrition and Bromatology and Research Institute for Food Analysis (I.I.A.A.), University of Santiago de Compostela, E-15782 Santiago de Compostela, Galicia, Spain

**Keywords:** lipid mediators, ω-3 and ω-6 PUFAs, non-targeted analysis, fragmentation patterns

## Abstract

A non-targeted strategy to simultaneously screen for over 100 lipid mediators from ω-6 ARA and ω-3 EPA and DHA fatty acids is presented. The method based on an extensive study of fragmentation patterns obtained by SPE-LC-MS/MS analysis-provided fingerprints to comprehensively elucidate and identify lipid mediators in biological samples. Many of these metabolites are associated to metabolic disorders, inflammatory, immune and oxidative stress. The methodology consisted of a three-step procedure. (1) SPE extraction of compounds from plasma and adipose tissue was followed by LC-MS/MS analysis operating in full scan mode. The methodology was validated for a group of 65 metabolites using standards. SPE recoveries ranged from 29–134% and matrix effect from 10–580%. LOD and LOQ ranged from 0.01 to 1765 ng/mL and 0.03 to 5884 ng/mL respectively, similarly than current analytical strategies based on MRM mode. (2) An extensive study of the mass spectra of a wide range of compounds was done to stablish a specific fragmentation pattern. Interestingly, illustrative fragmentations and new specific transitions to identify EPA and DHA lipid mediators have been innovatively established. (3) After analysis, 30 lipid mediators were tentatively identified in plasma and 35 in adipose tissue of rats according to the pre stablished fragmentation patterns. The hypothetical identification of compounds was validated by using reference standards. Around 85–90% of proposed identifications were correctly assigned and only 4 and 3 identifications failed in adipose tissue and plasma, respectively. The method allowed the identification of these metabolites without losing information by the use of predefined ions list. Therefore, the use of full scan mode together with the study of fragmentation patterns provided a novel and stronger analytical tool to study the complete profile of lipid mediators in biological samples than the analysis through MRM based methods. Importantly, no analytical standards were required at this qualitative screening stage and the performance and sensitivity of the assay were very similar to that of a MRM method.

## 1. Introduction

Lipid mediators have drawn much attention in clinical studies because they are a family of molecules that play a key role in the progression of inflammation and metabolic disorders [1]. They exhibit a large number of physiological effects in almost every organ system because of highly coordinated active processes including cellular signaling and transcription factors regulation, modulation of metabolic pathways and inflammation resolution, [2]. The production of these bioactive compounds is mediated by cyclooxygenases, lipoxygenases, cytochrome P450 mono- oxygenases, and non-enzymatic pathways [3] from the main ω-6 and ω-3 PUFAs as it is shown in Figure 1a,b. A very large number of isomeric compounds from different families can be produced. Specific examples include the families of PGs, IsoPs, TXs, and LTs which have been described as pro-inflammatory substances [1,4]. The most studied compounds derived from ARA are PGE_2_, PGF_1α_, PGF_2α_, TXA_2_ and 8iso-PGF_2α_, which have been associated with the chronic inflammatory stage of hypercholesterolemia, liver cirrhosis, myocardial reperfusion, type-2 diabetes, obesity, atherosclerosis, cancer, and CVD [5,6,7,8]. On the other hand, the corresponding metabolites produced from EPA and DHA have been suggested as less harmful compounds than the ω-6 ones [9,10]. Oxygenated lipid mediators derived from EPA and DHA, have been suggested one of the most efficient mechanisms to stop inflammation in a highly coordinated, active process [11]. Several works indicates that these mediators can resolve acute inflammatory responses with minimal damage to the surrounding tissue in a variety of inflammation situations [12,13]. Molecules like LXs [14], epoxides [15], and the recently discovered Rvs, PDs, and MaRs [16]. have been associated with the resolution of inflammation in CVD [6], arthritis [17], asthma [18], cancer [19,20], or Alzheimer [21,22].

The simultaneous analysis of the different families is a tough challenge, mainly because of their low physiological levels and the low stability [23], the matrix background interferences, and the large number of isomers with similar physicochemical properties and mass spectra that hinder their chromatography resolution and MS/MS identification [24]. Therefore, a highly sensitive and selective analytical procedure is essential to comprehensively study these lipid classes. Mass spectrometry has played a central role in quantitative assays for most of lipid mediators but also in their structural elucidation.

While some progress has been made on the decomposition behavior of ω-6 metabolites, there is a great deal more to understand ω-3 metabolism. For many years, GC-MS was the method of choice for lipids mediators [25]; nevertheless, the widespread LC-MS coupled with high-sensitivity ESI has provided a new approach for quantification, minimizing sample preparation requirements and particularly avoiding derivatization reactions [26]. Commonly, LC-MS/MS procedures follow MRM targeted analysis. Methods available in literature range from the simultaneously analysis of a few compounds [27,28] to more than one hundred [29,30,31]. Although these assays are sensitive and selective and cover a large number of metabolites, they are limited due to the trade-off between the decreases of sensitivity and the increase of the number of monitored compounds. In addition, MRM is limited to the analysis of a specific targeted group of compounds that operator must pre-define prior to the analysis. Therefore, the method is not suitable for a screening of the whole family of lipid mediators in biological samples unless a big number of expensive analytical standards are used to stablish the MS/MS transitions list. 

Non-targeted LC-MS/MS strategies register the full mass spectra of each group of compounds having the same parent mass. Then, this method favors the structural elucidation of metabolites enhancing the identification of co-eluting isomers. Moreover, this procedure allows the identification of many lipid mediators based on the study of fragmentation patterns without preselecting a group of compounds that might be or not in samples. Few works proposing screening methods of MS/MS have been found in literature to analyze lipid mediators. One of them was performed by Masoodi et al. [24] who validated an analytical methodology for the screening of lipid mediators combining the high mass resolution authority of an Orbitrap instrument and the study of MS/MS transitions. In our study, we achieved good identifications by using a LIT and suggested new specific transitions to identify lipid mediators.

The aim of this research was to develop a sensitive and selective non-targeted MS/MS method that allows the simultaneous screening of more than 100 lipid mediators in a single run. The procedure is based on the exhaustive interpretation of the mass spectra obtained. Firstly, SPE extraction of analytes from plasma and adipose tissue was carried out by and the procedure was validated for most of the different families of lipid mediators. 65 analytical standards of the representative metabolites were used for recovery experiments. Secondly, the mass spectra of the main ω-6 and ω-3 lipid mediators representative of the different families were exhaustive interpreted in order to systematize a pattern of diagnostic fragments for each compound. Then, a 24 min LC-MS/MS method operating in full scan mode and including the parent mass of the main families of compounds was optimized. Chromatographic gradient and mass spectrometry parameters were set for this representative group of 65 analytical standards. Finally, biological samples were analyzed and unknown peaks were tentatively identified matching specific MS/MS transitions with fragmentation patterns previously selected. Identifications were further validated by using analytical standards. This work provides an extensive interpretation and compilation of distinctive fragmentation patterns of lipid mediators that allows the use of non-targeted MS/MS strategies for the non-limited screening of biological samples.

## 2. Results and Discussion

### 2.1. SPE-LC-MS/MS Validation

The complete SPE-LC-MS/MS methodology applied for the extraction and analysis of lipid mediators. was validated with 65 standards representative of the main groups of compounds. The retention times, the selected MS/MS transitions, R^2^ linearity, repeatability, reproducibility, LODs, and LOQs of these 65 analytical standards are summarized in Table 1. 

The linear dynamic range of response was observed between 1 ng/mL and 1000 ng/mL for each compound. R^2^ values were higher than 0.99 for most compounds. LOD and LOQ values ranged between 0.01 to 17.65 ng/mL and 0.03 to 58.84 ng/mL, respectively, and these values were in agreement with those previously reported in the literature [16,28,32]. The instrumental intra-day repeatability and inter-day reproducibility at the low, intermediate and high concentration ranges (*n* = 4) showed acceptable precision, with values below 15 %RSD for almost all compounds.

Finally. Table 2 summarizes the repeatability of the entire optimized method, the global recovery, SPE recovery and matrix effects for spiked samples. The %RSD obtained in the different set of recovery experiments ranged between 0.01–21 in plasma and 0.4–67 in adipose tissue and they were slightly higher than instrumental %RSD. Thus, the precision of the entire protocol is mainly controlled by the sample extraction rather than by the mass analyzer. Global recoveries ranged between 7–489% in plasma and 18–442% in adipose tissue and recoveries of the SPE step were around 100 ± 15% for most of the compounds in plasma and were lower in adipose tissue, ranging between 40–115%. The main parameter that affected the global recovery of the method was the matrix effect. It ranged between 10–494% in plasma samples and between 21–580% in adipose tissue. There might be multiple reasons for the loss/gain of signal in the mass spectrometer. Among the main causes stand out the ion suppression and the overproduction of the [M-H]^-^ precursor ions of analytes in the ESI source due to co-eluting substances from the matrix. Matrix effects were considerable for some metabolites eluting in the last part of the chromatography.

### 2.2. Fragmentation Patterns Through the Analysis of Mass Spectra of Lipid Mediators

Here it is presented an extensive study of the MS/MS spectra of over 100 ARA. EPA and DHA lipid mediator that allowed determining specific fragmentation patterns useful for identification purposes. Most of MS/MS spectra and/or stereochemical structure of compounds were obtained from the LIPID MAPS database (https://www.lipidmaps.org). In the next paragraphs, fragmentation patterns for the different families of lipid mediators are discussed and specific cleavages aimed to identify the different chemical isomers are suggested. Most of the collision-induced decompositions suggested have been also discussed by previous published works. Specially, those compounds derived from the metabolism of ARA [33]. However, to the best of our knowledge, illustrative fragmentations of EPA and DHA derivatives some are innovatively proposed in this work. In addition to the study of mass spectra, the main difficulties in the identification of some isomers are also pointed out, as are common false positive identifications and so on.

#### 2.2.1. Monohydroxides

A common fragmentation pattern was observed for monohydroxides of ARA (HETEs), EPA (HEPEs) and DHA (HDoHEs), that was independent from the number of carbons of the PUFA precursor. MS/MS spectra showed common product ions of high intensity that involved the cleavage of the hydroxyl group (−18 *m/z* units = loss of H_2_O) and further loss of 44 *m/z* units (CO_2_) [34]. Murphy et al. [33] have suggested that ARA hydroxides produce a type of carbon-carbon cleavage with retention of the charge in the carboxyl group stabilized by resonance due to the high unsaturation of the molecules. These cleavages gave rise to several common product ions for all isomers. The specific transitions that allowed differentiating these HETEs originated from the cleavage of a carbon-carbon bond adjacent to the hydroxyl group (α-hydroxy cleavage) at the allylic position and a transfer of one proton to the unsaturated site [35]. We have suggested the same behavior for EPA and DHA hydroxides on the basis of the analyzed spectra and similarities with chemical structures of ARA hydroxides. The *m/z* of the resulting aldehyde and/or ketone depended on the position of the hydroxyl group and thus it allowed unambiguous determination of the regioisomers. The aldehyde also produced a specific fragment by a concomitant loss of 28 *m/z* units (loss of CO). The fragmentation pattern of 8-HETE, 12-HEPE, and 17-HDoHE as an example of hydroxides from each group are explained in detail in Figure 2. Similar patterns were found for the other studied regioisomers whose spectra were available at LIPID MAPS. A total of eight HETEs, seven HEPEs and ten HDoHEs were analyzed and the corresponding fragmentations are shown in Appendix A.

#### 2.2.2. Hydroperoxides

The spectra of HpETEs, HpEPEs and HpDoHEs (Figure 3 and Appendix A) showed a common pattern of fragmentation based on the highly reactivity of the functional group with nearby double bonds. The characteristic fragmentation suggested was very similar to the analogous hydroxides and independent from the number of carbons of the PUFA precursor. For regioisomers without spectra available in database, specific fragments were hypothesized based on the stoichiometric structure (Table 3). The loss of two molecules of H_2_O, CO_2_, and carbon-carbon bond cuts produced a sort of common fragments between isomers. The location of the hydroperoxy group in the chain determined the formation of characteristic fragments that allowed differentiating the isomers of each PUFA. As for corresponding hydroxides, the α-hydroxy cleavage of carbon-carbon bond resulted into an aldehyde with characteristic *m/z* and further loss of CO. Generally, most intense fragments were produced by the α-hydroxy cleavage with stabilization of the hydroperoxyl group in a diol structure, and then the loss of H_2_O and formation of an enol stabilized by resonance. A total of seven hydroperoxide spectra were studied and specific fragments were suggested for another seventeen isomers.

#### 2.2.3. Prostaglandins

The analysis of twelve spectra of type-2 prostaglandins from ARA and type-3 from EPA revealed similar decomposition pathway behavior. Non-specific cleavages that allowed distinguishing isomers were found according to the stereochemical structure of molecules that generally consists of a cyclopentane ring and two side chains with carboxylic and hydroxy groups, respectively. 

Intense product ions resulted from the subsequent losses up to three molecules of H_2_O and losses of CO_2_ from intermediate ions that were produced by the cleavage of the carboxylic group and stabilization of the charge by resonance of the hydroxyl with nearby double bonds. The carbon-carbon cleavage in the vinylic position of the C-15 hydroxyl produced a neutral hexenal compound by a proton transfer and several characteristic fragments with additional losses of H_2_O, CO_2_ and transfer of protons. An intermediate carbanion produced by the α-hydroxy cleavage may produce the rupture of the ring and stabilization by conjugate dienes of the corresponding fragment. These findings agree with previous published descriptions of the fragmentation of the prostaglandins family [35,36,37]. It is possible to unambiguously differentiate up to six groups of prostaglandins based on the parent mass *m/z*: 367, 353, 351, 349, 333 and 331. However, only a good chromatographic resolution of the peaks between isomers with the same *m/z* may lead to their differentiation due to their fragmentations are virtually identical. Figure 4 and Appendix A graphically explain some examples of prostaglandin fragmentation patterns.

#### 2.2.4. Resolvins, Protectins and Maresins

Recent experiments have uncovered that EPA and DHA serve as precursors for the production of potent bioactive mediators called resolvins, protectins and maresins [38]. The fragmentations of RvD_1_, RvE_1_ and PD_1_ have been described in detail in Figure 5. Common fragments were produced due to the higher unsaturation of the molecules and the number of hydroxy functional groups (up to 3) that gave rise to many losses of H_2_O and CO_2_. Specifically. the α-cleavage of hydroxyl determined several characteristic fragments as suggested by Serhan et al. [39]. Generally, the carbon-carbon bond was cleaved in the vinylic position when the hydroxyl was more proximal to the end of the carbon chain; however, it was cleaved in allylic position when proximal to the carboxylic group giving an aldehyde ion. Further losses of H_2_O and CO_2_ were produced from intermediate fragments too. Nevertheless, due to the novelty of these molecules, all isomers have not been widely characterized and spectra were not available, so we have suggested possible fragments for identification of series-D and E resolvins and MaR-1 based on the chemical structures and a similar behavior of the isomers in the decomposition pathways (Appendix A). The fragmentation proposed for MaR-1 was in agreement with the mass characterization of in vitro studies reported by Dalli et al. [40].

#### 2.2.5. Leukotrienes, Lipoxins and Dihydroxides

Another thirteen di-/trihydroxyeicosanoids derived from ARA and EPA have been studied and classified into four groups: LTs, LXs, DiHETEs and DiHETrEs. A common fragmentation pattern was observed, as described for resolvins and protectins, that produced common ions between isomers (losses of H_2_O and CO_2_). MS/MS spectra showed specific transitions depending on the location of hydroxy groups that were originated from α-hydroxy cleavages of adjacent carbon-carbon bonds and concomitant losses of H_2_O, CO_2._ and CO. The same trend as for resolvins and protectins was observed, with carbon-carbon cleavages generally occurring in the vinylic position of the hydroxyl located close to the terminal of the chain. On the contrary, the α-hydroxy cleavages proximal to the carboxylic group usually occurred in the allylic position. Results were in agreement with the MS/MS fragmentation pathways suggested by Murphy et al. [33] and Wheelan et al. [41].

Detailed decomposition pathways are shown in Figure 6 and Appendix A. The unambiguous identification of polyhydroxy compounds resulted difficult, due to the fact they may produce common fragments as other eicosanoids with the same parent mass ion. That is the case of DiHETEs and HpHETEs (*m/z* 335), LXA_4_-B_4_ and some TXs and PGs (*m/z* 351), LXA_5_ and resolvins and some prostaglandins (*m/z* 351). It is also important to note that DiHETEs may produce similar fragmentation patterns than derived HEPEs after the initial loss of H_2_O. However, the different parent mass allows distinguishing without doubts between these two groups of similar eicosanoids.

#### 2.2.6. Epoxy and Oxo-derivatives

As described by Murphy et al. [33], epoxy derivatives of ARA (EpETrEs) showed fragmentation patterns very close to those of HETEs and keto derivatives of ARA showed similar fragmentations as HEPEs. In the case of EpETrEs, the spectra generally resulted in a mixture of the two spectra from HETEs with functional groups in the same locations as the epoxy group (Figure 7 and Appendix A). Distinguishing some isomers from both isobaric groups—EpETrEs and HETEs—may be challenging if they are not chromatographically resolved because they have the same parent mass *m/z* and share most of the intense transitions. However, the epoxy LTA_4_ with one more unsaturation in the chain could be easily differentiated from HETEs and EpETrEs due to the lower *m/z* of the parent ion. The identification of some compounds from oxoETEs and HEPEs may result even more complicated without good chromatography resolution because they have the same parent mass and produce exactly the same MS/MS transitions. The main cleavages from ARA oxoETEs in Figure 8 are explained. Powell et al. [42] and MacMillan et al. [43] have also studied fragmentations from some oxo-derivatives of ARA. In a similar manner, the MS/MS spectra of several keto-PGs showed the same fragments as the corresponding PGs (Figure 9 and Appendix A). However, the different parent mass allowed the unambiguous identification between both families.

#### 2.2.7. Thromboxanes

Finally, the group of thromboxanes from ARA (series 2) and EPA (series 3) showed common losses of subsequent H_2_O molecules and CO_2._ Thromboxanes are eicosanoids structurally characterized by a six-member ring fused between C-8 and C-12 of the fatty acid. Margalit et al. [44] have suggested that the characteristic rupture of the ring becomes more interesting for identification of these compounds. This cleavage produces an unstable diol structure that easily loses a H_2_O molecule and derived to a more stable aldehyde [45]. The spectra and fragmentation patterns of TXB_3_ and TXB_2_ are shown in Figure 10 and Appendix A and characteristic cleavages are suggested for TXA_2_ which spectrum was not available.

### 2.3. Tentative Identifications in Samples and Validation

Once the extraction and determination procedures were validated, and fragmentation patterns were studied, we tentatively identified lipid mediators in samples of plasma and adipose tissue. Firstly, compounds were searched according to the *m/z* of parent ion and suggested specific fragmentations. For the highly unstable hydroperoxides, the ionization process provoked the loss of H_2_O in the source, then, the parent ion minus *m/z* 18 was also considered as the parent ion. As it was pointed out before, the large number of isomers makes it difficult to establish a single transition for every compound that leads to unambiguous identifications. A total of 35 compounds in adipose tissue and 30 in plasma were tentatively identified using the diagnostic fragmentation patterns.

Next, the identification was validated using a set of standards. Up to 27 hypothetical identifications were validated in adipose tissue and 25 in plasma. Results showed a confirmation rate of our hypotheses of 85 and 88% in adipose tissue and plasma, respectively (4 and 3 proposed identifications failed, respectively).

Specifically, the presence of nine isomers in adipose tissue and seven in plasma from the family of HEPEs and oxoETEs was observed on the basis of the screening analysis of *m/z* 317. We tentatively identified 18-HEPE, 15-HEPE, 12-HEPE, 11-HEPE, 5-HEPE and 12-oxoETE in both samples and their presence was confirmed by using commercial standards. Specific and common cleavages were found for these compounds according to the rules described above. In addition, an increase in the retention time of hydroxides was observed when the position of hydroxyl was more proximal to the carboxylic group. This fact may be explained by a decrease in the polarity of the molecule. However, there were exceptions to this rule between 11-HEPE and 12-HEPE, 11-HETE and 12-HETE and 13-HDoHE and 14-HDoHE in agreement with Massey et al. [1]. 9-HEPE was suggested in adipose tissue although it was not validated. The isomers 5-oxoETE in both samples and 15-oxoETE in adipose were suggested. However, validation experiments showed non-coincidence of the retention times between standards and the hypothetical peaks.

Similarly, the presence of 15-HETE, 12-HETE, 11-HETE and 14.15-EpETE in both samples was suggested based on the screening analysis of *m/z* 319. 11.12-EpETE was only suggested in adipose tissue and 20-HETE in plasma. In addition, one peak was found in both samples leading to theoretical identification of 5.6-EpETE or 5-HETE. They cannot be unambiguously identified without retention time information due to similarities in the mass spectra and specific transitions. The use of 5-HETE standard confirmed the absence of this compound in samples. Additionally, it was observed that epoxy and hydroxy isomers with concurrent positions of functional groups did not co-elute under our LC conditions. Therefore a peak attributed either to 5-HETE or 5.6-EpETE was tentatively identified and then validated as the epoxy isomer.

Special attention was required for hydroperoxides from DHA (HpDoHE) (*m/z* 359), EPA (HpEPE) (*m/z* 333), and ARA (HpETE) (*m/z* 335). These compounds were found to display higher sensitivity using the ratios *m/z* 341, *m/z* 315 and *m/z* 317 as parent mass filter, respectively. In spite of the low-energy ionization used, hydroperoxides may lose a H_2_O molecule in the ESI source due to the low stability of the hydroperoxy group. Therefore, specific transitions for these compounds were searched for in the chromatogramf using both “parent mass” ratios. We could not confirm the hypothesis of the presence of HpDoHEs and HpETEs based on common and specific transitions. However, the use of commercial 17-HpDoHE showed its presence in both samples although it had not been previously hypothesized. The presence of 12-HpEPE was suggested and further validated in plasma.

Regarding DHA hydroxides, the analysis of *m/z* 343 showed a hypothetical identification of eight isomers in adipose tissue and six isomers in plasma based on specific transitions. Standards validation confirmed four hydroxides in adipose and three in plasma. No false identifications were seen in this group. In the same way. 335 *m/z* was analyzed searching for DiHETEs from ARA. Six isomers were tentatively found in adipose tissue and five in plasma, from which two were confirmed and one rejected by using standards. Then, the presence of PGE_2_/PGD_2_, ARA, EPA and DHA was suggested in both samples by analyzing *m/z* 351, 303, 301 and 327 parent mass ions, respectively, and the hypotheses were confirmed. Finally, 14.15-DiHETrE was hypothesized in adipose tissue based on 337 *m/z* screening and it was validated.

Seven other *m/z* ratios were also detected but their fragmentation pattern could not be assigned to any of the groups studied (624, 375, 369, 367, 353, 349 and 331). Results for tentative identifications and validations are summarized in Table 4, along with information about specific transitions.

## 3. Experimental Section

### 3.1. Chemicals

Thromboxane B_3_ (TXB_3_), prostaglandin D_3_ (PGD_3_), prostaglandin E_3_ (PGE_3_), prostaglandin D_2_ (PGD_2_), prostaglandin E_2_ (PGE_2_), prostaglandin B_2_ (PGB_2_), prostaglandin J_2_ (PGJ_2_), resolvin D_1_ (RvD_1_), resolvin D_2_ (RvD_2_), maresin R_1_ (MaR_1_), protectin D_x_ (PD_x_, 10(*S*),17(*S*)-dihydroxy-4*Z*,7*Z*,11*E*,13*Z*,15*E*,19*Z*-docosahexaenoic acid), leukotriene B_4_ (LTB_4_), leukotriene C_4_ (LTC_4_), isoprostane F_2α_ (8isoPGF_2α_), isoprostane F_3α_ (8isoPGF_3α_), 15-HpEPE (15(*S*)-hydroperoxy-5*Z*,8*Z*,11*Z*,13*E*,17*Z*-eicosapentaenoic acid), 12-HpEPE (12(*S*)-hydroperoxy- 5*Z*,8*Z*,10*E*,14*Z*,17*Z*-eicosapentaenoic acid), 15-HpETE (15(*S*)-hydroperoxy-5*Z*,8*Z*,11*Z*,13*E*- eicosatetraenoic acid), 12-HpETE (±12-hydroperoxy-5*Z*,8*Z*,10*E*,14*Z*-eicosatetraenoic acid), 5-HpETE (5(*S*)-hydroperoxy-6*E*,8*Z*,11*Z*,14*Z*-eicosatetraenoic acid), 17-HpDoHE (17(*S*)-hydroperoxy- 4*Z*,7*Z*,10*Z*,13*Z*,15*E*,19*Z*-docosahexaenoic acid), 18-HEPE (±18-hydroxy-5*Z*,8*Z*,11*Z*,14*Z*,16*E*-eicosa- pentaenoic acid), 15-HEPE (±15-hydroxy-5*Z*,8*Z*,11*Z*,13*E*,17*Z*-eicosapentaenoic acid), 12-HEPE (±12-hydroxy-5*Z*,8*Z*,10*E*,14*Z*,17*Z*-eicosapentaenoic acid), 11-HEPE (±11-hydroxy-5*Z*,8*Z*,12*E*,14*Z*, 17*Z*-eicosapentaenoic acid), 5-HEPE (±5-hydroxy-6*E*,8*Z*,11*Z*,14*Z*,17*Z*-eicosapentaenoic acid), 17,18-DiHETE (±17,18-dihydroxy-5*Z*,8*Z*,11*Z*,14*Z*-eicosatetraenoic acid), 17,18-EpETE (±17,18-epoxy- 5*Z*,8*Z*,11*Z*,14*Z*-eicosatetraenoic acid), 11,12-EpETE (±11,12-epoxy-5Z,8Z,14Z,17Z-eicosatetraenoic acid), 14,15-EpETE (±14,15-epoxy-5*Z*,8*Z*,11*Z*,17*Z*-eicosatetraenoic acid), 8,9-EpETE (±8,9-epoxy- 5*Z*,11*Z*,14*Z*,17*Z*-eicosatetraenoic acid), 20-HETE (±20-hydroxy-5*Z*,8*Z*,11*Z*,14*Z*-eicosatetraenoic acid), 15-HETE (±15-hydroxy-5*Z*,8*Z*,11*Z*,13*E*-eicosatetraenoic acid), 12-HETE (±12-hydroxy-5*Z*,8*Z*,10*E*, 14*Z*-eicosatetraenoic acid), 11-HETE (±11-hydroxy-5*Z*,8*Z*,12*E*,14*Z*-eicosatetraenoic acid), 5-HETE (±5-hydroxy-6*E*,8*Z*,11*Z*,14*Z*-eicosatetraenoic acid), 14,15-DiHETE (±14,15-dihydroxy-5*Z*,8*Z*,11*Z*, 17*Z*-eicosatetraenoic acid), 5,15-DiHETE (5(*S*),15(*S*)-dihydroxy-6*E*,8*Z*,10*Z*,13*E*-eicosatetraenoic acid), 5,6-DiHETE (±5,6-dihydroxy-8*Z*,11*Z*,14*Z*,17*Z*-eicosatetraenoic acid), 11,12-DiHETrE (±11,12- dihydroxy-5*Z*,8*Z*,14*Z*-eicosatrienoic acid), 11,12-EpETrE (±11,12-epoxy-5*Z*,8*Z*,14*Z*-eicosatrienoic acid), 8,9-DiHETrE (±8,9-dihydroxy-5*Z*,11*Z*,14*Z*-eicosatrienoic acid), 8,9-EpETrE (±8,9-epoxy- 5*Z*,11*Z*,14*Z*-eicosatrienoic acid), 14,15-DiHETrE (±14,15-dihydroxy-5*Z*,8*Z*,11*Z*-eicosatrienoic acid), 14,15-EpETrE (±14,15-epoxy-5*Z*,8*Z*,11*Z*-eicosatrienoic acid), 5,6-DiHETrE (±5,6-dihydroxy-8*Z*,11*Z*, 14*Z*-eicosatrienoic acid), 5,6-EpETrE (±5,6-epoxy-8*Z*,11*Z*,14*Z*-eicosatrienoic acid), 17-HDoHE (±17-hydroxy-4*Z*,7*Z*,10*Z*,13*Z*,15*E*,19*Z*-docosahexaenoic acid), 14-HDoHE (14(S)-hydroxy-4*Z*,7*Z*, 10*Z*,12*E*,16*Z*,19*Z*-docosahexaenoic acid), 11-HDoHE (±11-hydroxy-4*Z*,7*Z*,9*E*,13*Z*,16*Z*,19*Z*- docosahexaenoic acid), 4-HDoHE (±4-hydroxy-5*E*,7*Z*,10*Z*,13*Z*,16*Z*,19*Z*-docosahexaenoic acid), 19,20-DiHDPA (±19,20-dihydroxy-4*Z*,7*Z*,10*Z*,13*Z*,16*Z*-docosapentaenoic acid), 19,20-EDP (±19,20- epoxy-4*Z*,7*Z*,10*Z*,13*Z*,16*Z*-docosapentaenoic acid), 7(*S*)-17(*S*)dihydroxy-8*E*,10*Z*,13*Z*,15*E*,19*Z*- docosapentaenoic acid), 10,11-EDP (±10,11-epoxy-4*Z*,7*Z*,13*Z*,16*Z*,19*Z*-docosapentaenoic acid), 16,17-EDP (±16,17-epoxy-4*Z*,7*Z*,10*Z*,13*Z*,19*Z*-docosapentaenoic acid), 7,8-EDP (±7,8-epoxy-4*Z*,10*Z*, 13*Z*,16*Z*,19*Z*-docosapentaenoic acid), 13,14-EDP (±13,14-epoxy-4*Z*,7*Z*,10*Z*,16*Z*,19*Z*-docosa- pentaenoic acid), 15-oxoETE (15-oxo-5*Z*,8*Z*,11*Z*,13*E*-eicosatetraenoic acid), 12oxoETE (12-oxo-5*Z*,8*Z*,10*E*,14*Z*-eicosatetraenoic acid), 5-oxoETE (5-oxo-6*E*,8*Z*,11*Z*,14*Z*-eicosatetraenoic acid), 12-HETE-d_8_ (12(*S*)-hydroxy-5*Z*,8*Z*,10*E*,14*Z*-eicosatetraenoic-5,6,8,9,11,12,14,15-d_8_ acid), EPA (5*Z*,8*Z*,11*Z*,14*Z*,17*Z*-eicosapentaenoic acid), DHA (4*Z*,7*Z*,10*Z*,13*Z*,16*Z*,19*Z*-docosahexaenoic acid), and ARA (5*Z*,8*Z*,11*Z*,14*Z*-eicosatetraenoic acid) were purchased from Cayman Chemicals (Ann Arbor, MI, USA). The supplier stated purities higher than 96% for all standards. Methanol and water, Optima LC-MS grade, were purchased from Fisher Scientific (Branchburg, NJ, USA); methyl formate was purchased from Sigma Aldrich (Poole, UK); n-Hexane was provided by Merck (Darmstadt, Germany) and ethanol, formic acid and hydrochloric acid were from AnalR Normapur (Fontenai, France). SPE cartridges (Oasis-HLB, 60 mg, 3 mL) were supplied by Waters (Milford, MA, USA).

### 3.2. Plasma and Adipose Tissue Samples

Blood from Wistar rats from a previous experiment was collected and centrifuged at 850 g (4 °C, 15 min) in the presence of EDTA to remove erythrocytes. Then, plasma was supplemented with 5 mM phenyl methyl sulfonyl fluoride (PMSF, protease inhibitor) and erythrocyte free samples were immediately stored at −80 °C until use. The defrosting process was carried out slowly in darkness, to prevent possible analyte oxidation and degradation [46]. Total abdominal fat, corresponding to the sum of epididymal, perirenal and mesenteric depots was excised, washed with 0.9% NaCl solution, weighed and immediately frozen in liquid nitrogen upon sacrifice.

### 3.3. Sample Preparation by Solid Phase Extraction

Lipid mediators from plasma samples were extracted following a modified SPE method previously described [47]. Briefly, 1300 µL of cold methanol: water (30:70, *v*/*v*) was used to extract metabolites from 200 µL of spiked plasma with internal standard 12-HETE-d8 (500 ng/mL). After centrifugation, remaining solution was loaded into conditioned Oasis-HLB cartridges (60 mg, 3 mL, Waters, Milford, MA, USA) with 5 mL methanol (0.5% BHT) and 5 mL water in succession. Then, cartridges were washed with 5 mL cold water, 5 mL cold methanol: water (15:75, *v*/*v*), and 2.5 mL cold hexane in succession. After sorbent dryness, compounds were eluted with 2 mL of cold methyl formate (0.1% BHT). Extracts were evaporated to dryness in a speed-Vacuum system from Supelco (Bellefonte, PA, USA) coupled to a vacuum pump from Millipore (Bedford, MA, USA). The residue was dissolved in 30 µL cold ethanol and stored at −80 °C prior to LC-MS/MS analysis.

Metabolites from adipose tissue were extracted using a modified method of a published protocol [48]. 150 mg of frozen tissues were cut, spiked with internal standard 12-HETE-d8 and extracted by adding 1 mL cold-methanol (0.5% BHT) and sonication for 1min under 0.6 sec cycle and 100% of amplitude (Labsonic sonicator from Sartorius, city, Germany). Samples were incubated on ice for 10 min and then centrifuged at 800 g for 10 min, at 4 °C, to remove potential proteins that may cause interference. Supernatant was transferred to a clean amber glass vial and pellet was subjected to a second extraction in methanol and sonication as described. Both supernatants were collected together and final pellet was washed with 500 µL cold methanol: water (30:70, *v*/*v*) twice. The pooled fractions were diluted with 4.6 mL of cold water to a final solution of methanol: water (30:70, *v*/*v*). Then, samples were loaded to SPE cartridges following the same procedure as plasma samples.

### 3.4. LC-MS/MS

Chromatographic separation was carried out on a Dionex UltiMate 3000 Series (Thermo Fisher, Rockford, IL, USA) that includes a binary pump, a degasser system and a thermostated autosampler. Compounds were separated on a C18-Symmetry column, 150 × 2.1 mm, 3.5 μm (Waters) protected with a 4 × 2mm C18 guard cartridge provided by Phenomenex (Torrance, CA, USA). A binary eluent system of water (A) and methanol (B), both with 0.02% (*v*/*v*) of formic acid, was used as mobile phase. The gradient was: 0–1 min (60% B), 2–12 min (80% B), 13–18 min (100% B), and 19–24 min (60% B). The flow rate was set to 0.2 mL/min, the column effluent was directly introduced in the ESI interface without splitting, and injection volume was set to 10 μL. The column was maintained at room temperature and extracts were kept at 4 °C in the autosampler. Mass spectrometry analyses were carried out on a dual-pressure linear ion trap LTQ Velos Pro (Thermo Fisher). Operating conditions of the ESI source were negative ion mode with a sheath gas flow rate of 40 units, spray voltage of 5.5 kV, capillary temperature of 300 °C and S-lens radio-frequency level of 60%. Nitrogen was used as nebulizing gas and helium was the collision gas. Instrument control and data acquisition were done with Xcalibur software. Compounds were divided into 22 groups according to the *m/z* of their parent ion. Acquisition was set as full scan mode ranged from 90 to 400 *m/z* units with the 22 parent masses in a single segment during the whole run. The isolation width was set at 1. Collision energies for each *m/z* are summarized in Table 5 and they were set as a compromise among the most common used in literature and optimized collision energies for 65 analytical standards used for validation purposes.

### 3.5. Method Validation: Linearity. Repeatability. Reproducibility. LOD. LOQ. Recovery Experiments. and Matrix Effect

Eight batches of calibration solutions were used to determine the dynamic linear range of response for each of the 65 analytical standards in the range between 1 and 1000 ng/mL, with the exception of ARA. EPA and DHA, whose concentration in biological samples is higher, and thus linearity was evaluated in the range between 0.1 and 50 µg/mL. The calibration curves were calculated using least-squares linear regression. The internal standard (12-HETE-d_8_) was included in all calibration solutions at 500 ng/mL. The reproducibility (inter-day) and repeatability (intra-day) of the instrument was evaluated at three levels of concentration (*n* = 4) for each compound: 1, 50, and 1000 ng/mL; and 0.1, 10, 50 µg/mL for PUFAs, LODs and LOQs were estimated by using signal-to-noise (S/N) ratios of 3 and 10, respectively.

Different recovery experiments were performed to determine global recovery, SPE recovery and matrix effects: (a) The percentage of global recovery was evaluated comparing the signal for each metabolite between: spiked plasma and adipose tissue samples before preparation and a pool of analytical standards at the same level of concentration. (b) The percentage of SPE recovery was determined comparing samples spiked before preparation with samples spiked after the SPE step. (c) Matrix effect was evaluated comparing samples spiked after the SPE extraction with the pool of analytical standards. In any case, when metabolites were already present in blank samples (non-spiked samples), the area of chromatographic peaks in spiked samples was subtracted with the area in blanks. Experiments were done in triplicate and the repeatability of the entire method was expressed as percentage of relative standard deviation (%RSD).

### 3.6. Qualitative Screening of Samples and Validation

Plasma and adipose tissue were subjected to SPE-LC-MS/MS. Chromatographic peaks for each parent mass were extracted from MS^2^ selecting quantitative and selective transitions for each lipid mediator according to the fragmentation pattern previously studied. The spectra of each chromatographic peak was studied and compared with the fragmentation pattern to tentatively identify compounds in samples. Then, other samples were spiked before extraction with a pool of 65 standards described in the chemical section. Spiked samples were analyzed under the same LC-MS/MS conditions than non-spiked ones and hypothetical identifications were validated using the information of retention times and fragmentation in standards and samples.

### 3.7. Data Analysis

Means, standard deviations, least-squares linear regression curves, and percentages used to validate methodology were performed by using the free R software (version 3.2.4).

## 4. Conclusions

An extensive study of fragmentation patterns provided fingerprints allowing us to comprehensively elucidate and identify lipid mediators in biological samples obtained by SPE-LC-MS/MS analysis. It has allowed the identification of over 100 bioactive ω-6 and ω-3 metabolites. Interestingly, illustrative fragmentations of EPA and DHA derivatives have been innovatively established for successful identification. The fragmentation patterns study could be used for developing software in order to facilitate the identification of this big family of metabolites without the need of standards for qualitative purposes. The screening method was useful to correctly identify more than 85% of the compounds found in biological samples using the fragmentation patterns here proposed. Moreover, the method allowed the identification of these metabolites without losing information by the use of predefined ions list. Few works proposing MS/MS screening methods to analyze lipid mediators have been found in literature. In our study, we suggested new specific transitions to identify lipid mediators, especially for those derived from EPA and DHA and we achieved a high success score in identifications by using a LIT without the high mass resolution of an Orbitrap. Further improvements of these techniques are needed in order to increase the number of target metabolites that can be accurately identified in a single screening step before the quantitative analysis.

## Figures and Tables

**Figure 1 molecules-24-02276-f001:**
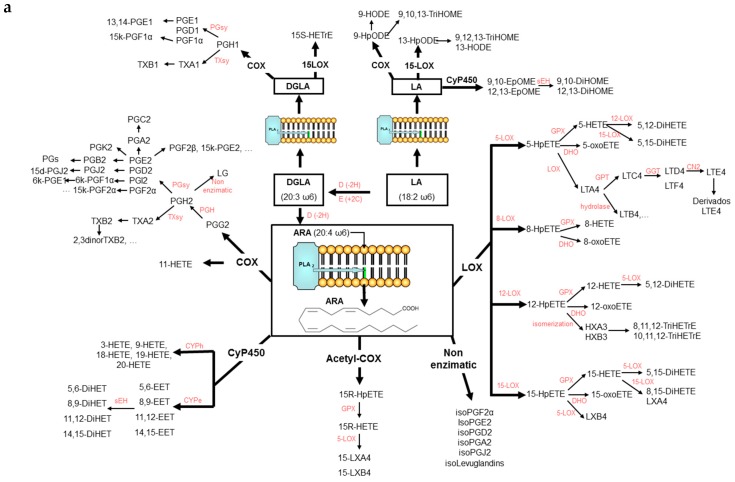
(**a**): Cascade of formation of lipid mediators from ω-6 LA, DGLA and ARA. (**b**): Cascade of formation of lipid mediators from ω-3 ALA, EPA, and DHA.

**Figure 2 molecules-24-02276-f002:**
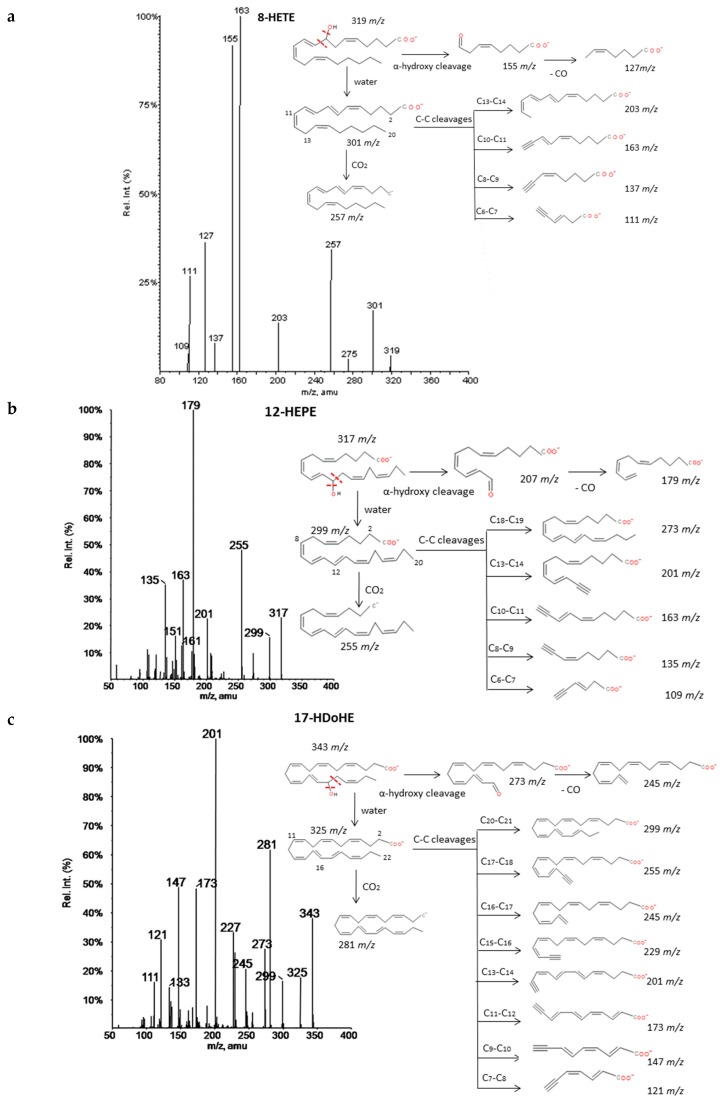
Fragmentation patterns suggested for (**a**) ARA, (**b**) EPA and (**c**) DHA hydroxides. Common cleavages and specific transitions for 8-HETE, 12-HEPE and 17-HDoHE. Product ion spectra at *m/z* 319 for 8HETE, *m/z* 317 for 12HEPE, and *m/z* 343 for 17HDoHE were obtained from the LIPID MAPS database at CID 30V.

**Figure 3 molecules-24-02276-f003:**
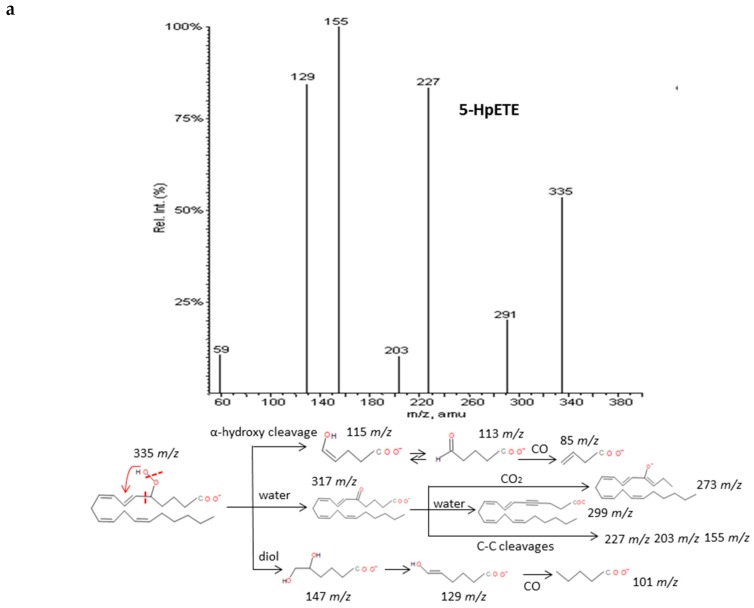
Fragmentation pattern suggested for (**a**) ARA, (**b**) EPA and (**c**) DHA hydroperoxides. Common cleavages and specific transitions for 5-HpETE, 5-HpEPE, and 17-HpDoHE. Product ion spectra at *m/z* 335 for 5-HpETE, *m/z* 333 for 15-HpEPE, and *m/z* 359 for 17-HpDoHE were obtained from the LIPID MAPS database at CID 30V.

**Figure 4 molecules-24-02276-f004:**
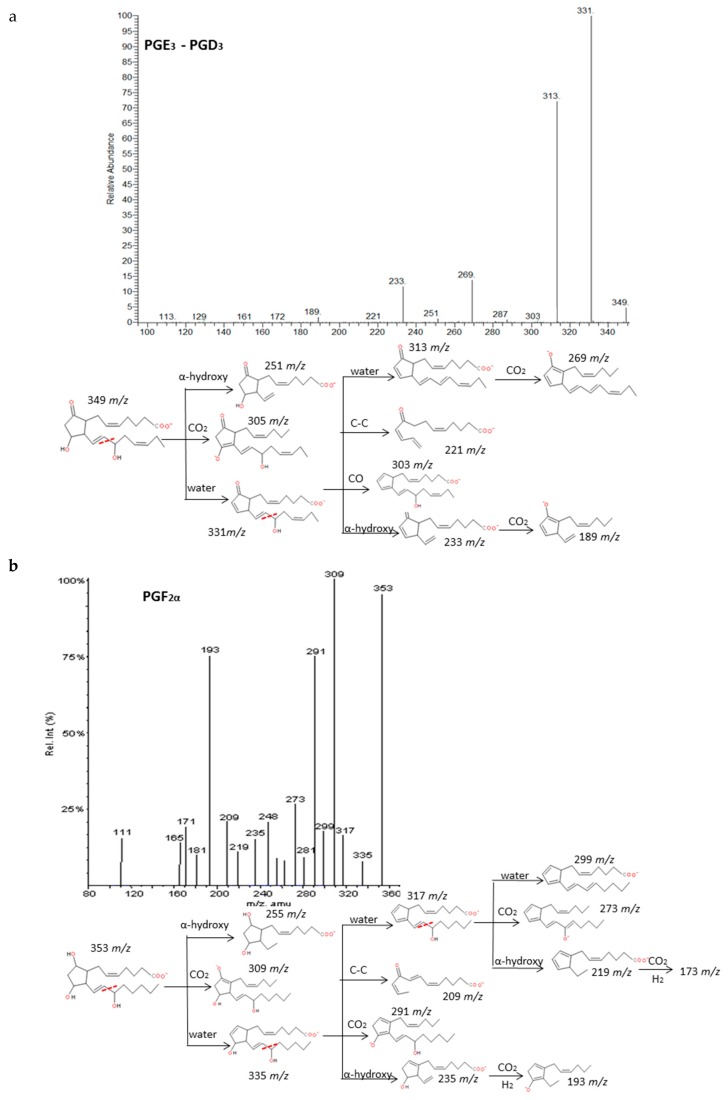
Fragmentation patterns suggested for (**a**) PGE_3_, PGD_3_, (**b**) PGF_2α._ and (**c**) PGE_2_. Product ion spectra at *m/z* 349, *m/z* 353, and *m/z* 351 were obtained from the LIPID MAPS database at CID 30V. An identical spectrum was found for PGE_3_ and PGD_3._

**Figure 5 molecules-24-02276-f005:**
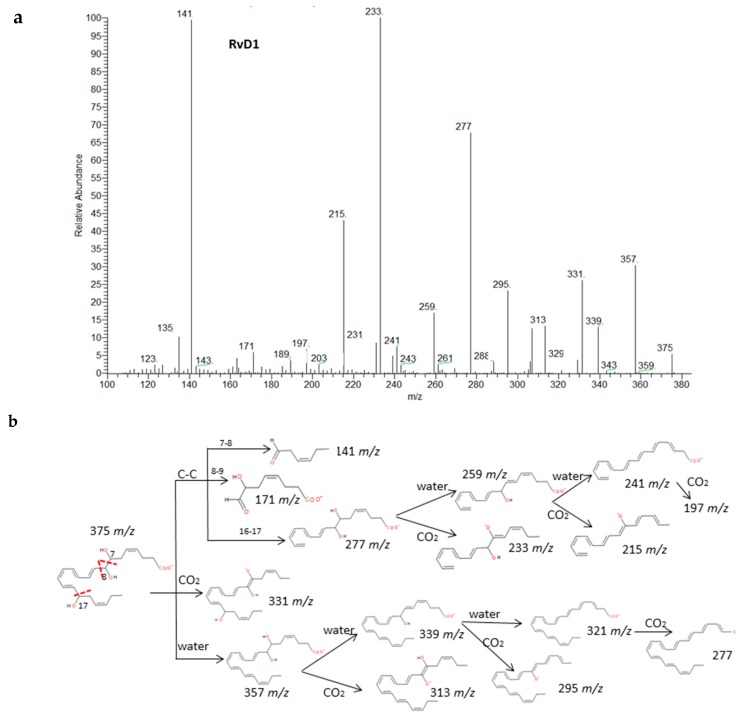
Fragmentation patterns suggested for (**a**,**b**) RvD_1_, (**c**) RvE_1._ and (**d**,**e**) PD_1_. Product ion spectra at *m/z* 375, *m/z* 349 and *m/z* 359 were obtained from the LIPID MAPS database at CID 30V.

**Figure 6 molecules-24-02276-f006:**
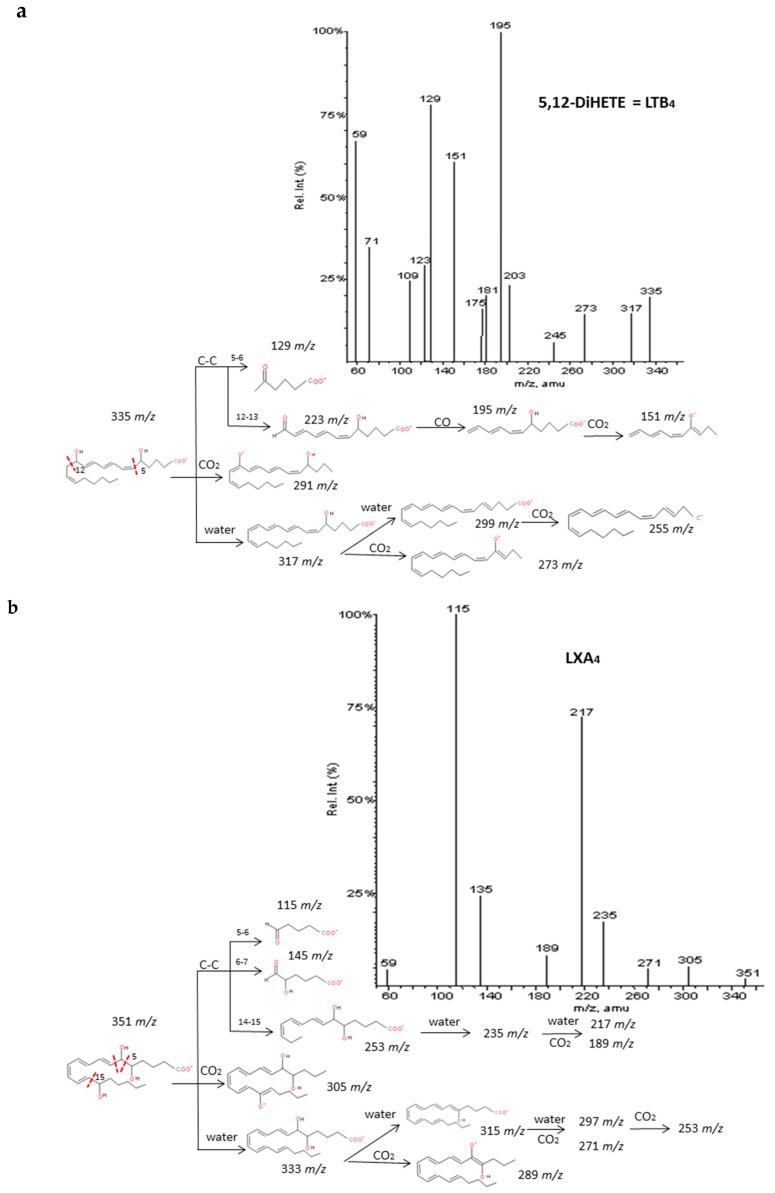
Fragmentation pattern suggested for (**a**) LxA_4_, (**b**) LTB_4._ and (**c**) 8.9-DiHETrE. Product ion spectrum at *m/z* 351. *m/z* 335. and *m/z* 337 were obtained from database of LIPID MAPSat CID 30V.

**Figure 7 molecules-24-02276-f007:**
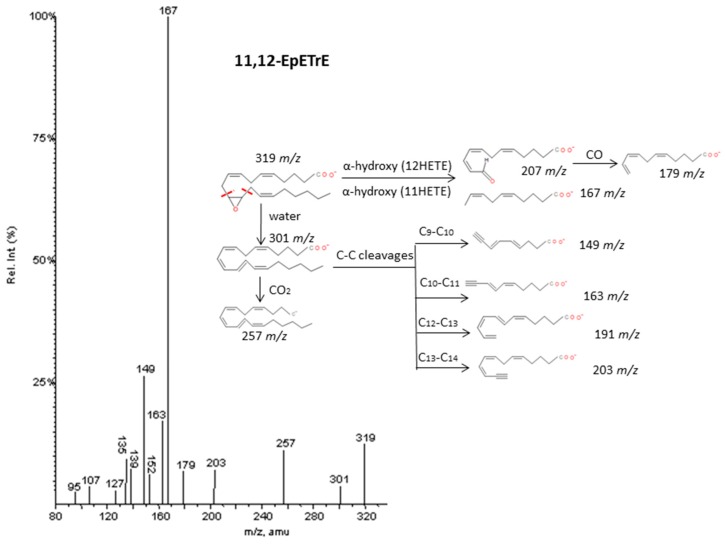
Fragmentation pattern suggested for 11.12-EpETrE. The product ion spectrum at m/z 319 was obtained from database of LIPID MAPS at CID 30V.

**Figure 8 molecules-24-02276-f008:**
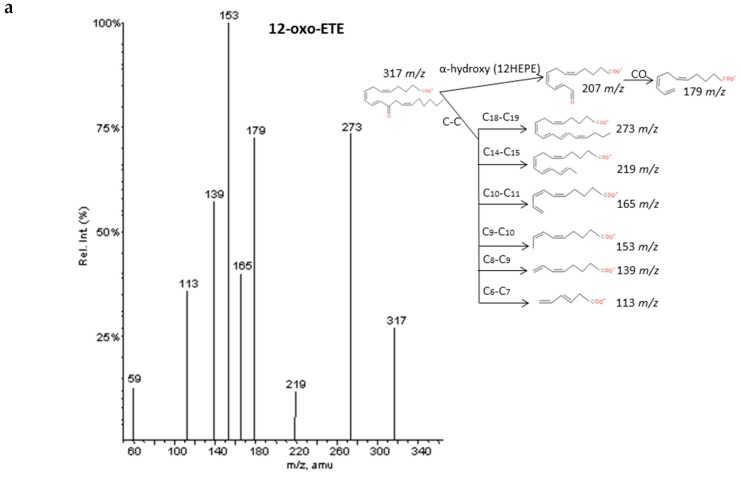
Fragmentation pattern suggested for (**a**) 12-oxoETE, (**b**) 5-oxoETE, and (**c**) 15-oxoETE (product ion spectrum at *m/z* 317 obtained from LIPID MAPS at CID 30V). A MS/MS spectrum was not found for 15oxoETE, thus, potential fragments for identification were suggested based on lipid structure.

**Figure 9 molecules-24-02276-f009:**
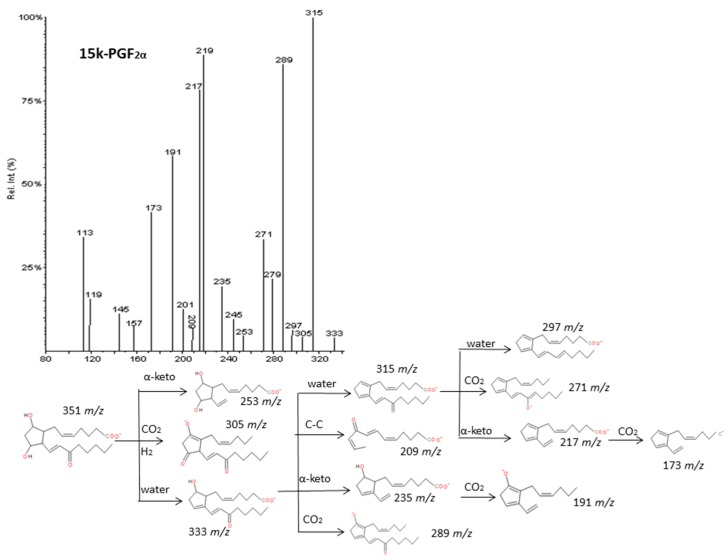
Fragmentation pattern suggested for 15k-PGF_2α_. The product ion spectrum at *m/z* 351 was obtained from the LIPID MAPS database at CID 30V.

**Figure 10 molecules-24-02276-f010:**
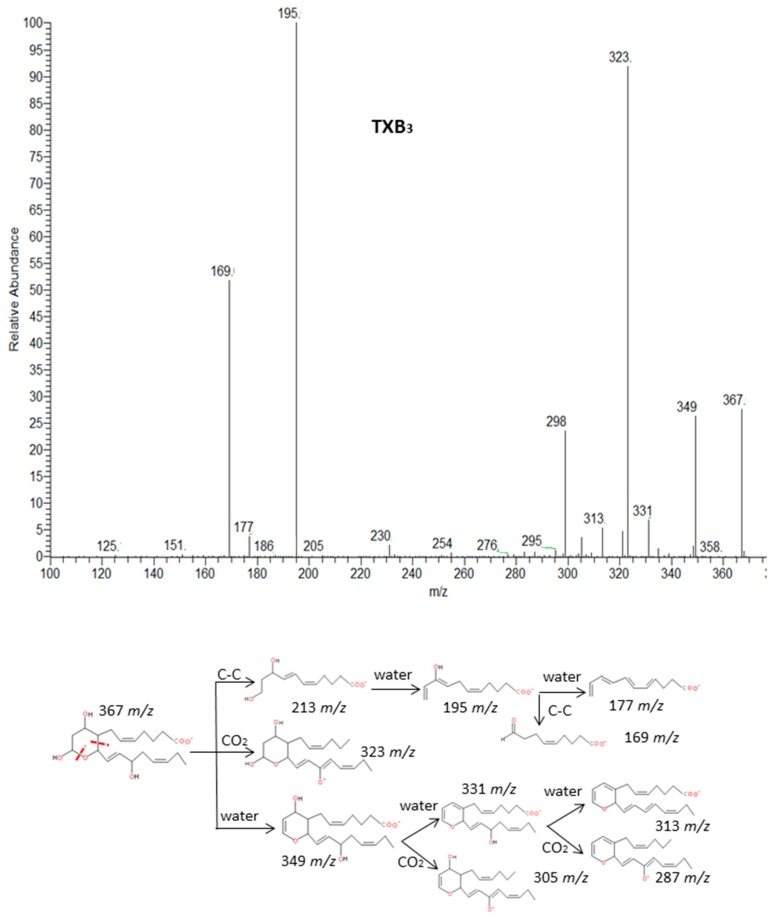
Fragmentation pattern suggested for TXB_3_. The product ion spectrum at *m/z* 367 was obtained from LIPID MAPS at CID 30V.

**Table 1 molecules-24-02276-t001:** Retention times. MS/MS quantification transitions (*m/z*). Linearity by least-squares linear regression (1 to 1000 ng/mL except for PUFAs: 0.1 to 50 µg/mL). Intra-day repeatability (*n* = 4) and inter-day reproducibility (*n* = 4) of LC-MS assays at three concentration levels (1-50-1000 ng/mL except for PUFAs: 0.1-10-50 µg/mL). LOD and LOQ of the 65 analytical standards.

Compound	Retention Time (min)	MS/MS Quantification	Linearity (R^2^)	Repeatability (%RSD)			Reproducibility (%RSD)			LOD (ng/mL)	LOQ (ng/mL)
				1	50	1000	1	50	1000		
15-HEPE	11.96	317 > 247	0.9996	1	1	3	11	1	3	0.37	1.24
18-HEPE	11.55	317 > 259	0.9999	7	1	1	7	2	3	0.18	0.58
12-HEPE	12.62	317 > 208	1.000	3	3	3	6	3	7	0.01	0.03
5-HEPE	14.17	317 > 115	0.9999	3	0	2	7	2	8	0.37	1.25
11-HEPE	12.13	317 > 195	0.9995	8	0	1	11	1	6	0.13	0.43
11-HETE	15.32	319 > 167	0.9968	4	4	8	18	8	9	0.06	0.21
5-HETE	15.96	319 > 115	0.9807	13	9	2	14	7	2	0.07	0.24
12-HETE	15.48	319 > 179	0.9991	4	3	7	7	7	8	0.03	0.11
15-HETE	14.83	319 > 219	0.9962	12	2	4	11	10	4	0.07	0.25
20-HETE	13.69	319 > 245	0.9988	4	2	2	11	4	4	0.53	1.76
4-HDoHE	16.08	343 > 281	0.9997	2	10	9	14	8	9	0.04	0.15
17-HDoHE	15.19	343 > 245	0.9998	5	3	2	10	4	5	0.29	0.96
14-HDoHE	15.43	343 > 205	0.9907	3	6	5	6	10	8	0.10	0.33
11-HDoHE	15.67	343 > 149	0.9985	10	6	6	9	8	5	0.07	0.25
15-OxoETE	13.69	317 > 113	0.9999	13	3	1	13	8	2	1.58	5.27
12-OxoETE	14.83	317 > 153	0.9944	nd	3	3	nd	71	27	1.21	4.03
5-OxoETE	15.95	317 > 203	0.9974	7	9	5	14	8	4	0.11	0.38
15-HpEPE	11.93	333/315 > 271	0.9986	nd	2	2	nd	14	3	0.59	1.95
12-HpEPE	12.42	333/315 > 271	0.9996	10	1	1	10	23	9	0.29	0.95
12-HpETE	15.39	335/317 > 153	0.9998	15	0	3	15	29	3	0.50	1.65
5HpETE	15.86	335/317 > 155	0.9620	nd	13	7	nd	16	9	6.48	21.59
15-HpETE	14.83	335/317 > 113	0.9991	nd	3	1	nd	9	1	17.65	58.84
17-HpDoHE	15.12	359/341 > 297	0.9998	13	1	3	11	5	3	1.05	3.51
14.15-DiHETE	8.58	335 > 207	0.9998	9	1	1	12	1	1	0.53	1.78
5.15-DiHETE	7.93	335 > 235	0.9983	10	0	5	15	17	8	0.18	0.61
5.6-DiHETE	10.56	335 > 145	0.9994	1	3	2	1	9	5	0.95	3.16
11.12-DiHETrE	10.68	337 > 167	0.9988	3	1	3	8	3	2	0.05	0.16
8.9-DiHETrE	11.66	337 > 185	0.9999	5	2	1	7	4	8	0.17	0.55
14.15-DiHETrE	10.03	337 > 207	0.9989	7	2	4	19	3	4	0.12	0.39
5.6-DiHETrE	13.87	337 > 145	0.9988	5	3	2	7	4	3	0.10	0.35
19.20-DiHDPA	10.60	361 > 273	0.9996	5	0	1	9	4	4	0.17	0.55
17.18-DiHETE	8.50	335 > 247	0.9989	2	3	2	13	6	4	0.46	1.54
7.17-hydroxiDPA	8.22	361 > 263	1.000	14	2	3	14	11	9	0.43	1.45
11.12-EpETrE	16.12	319 > 179	0.9890	13	9	11	11	7	9	0.17	0.58
8.9-EpETrE	16.20	319 > 155	0.9516	12	12	11	23	20	9	0.13	0.42
14.15-EpETrE	15.80	319 > 219	0.9920	11	15	7	11	13	6	0.26	0.87
5.6-EpETrE	16.28	319 > 191	0.9950	15	12	5	15	65	29	0.41	1.38
17.18-EpETE	13.85	317 > 259	0.9999	2	1	6	11	8	9	0.19	0.63
11.12-EpETE	15.07	317 > 179	0.9969	4	3	7	9	6	6	0.24	0.81
14.15-EpETE	14.83	317 > 207	0.9987	6	2	4	15	12	11	0.27	0.91
8.9-EpETE	15.32	317 > 155	0.9998	19	3	6	13	5	5	0.50	1.67
19.20-EDP	15.84	343 > 241	0.9945	11	5	6	18	10	14	0.30	0.99
10.11-EDP	16.08	343 > 153	0.9976	2	6	8	10	19	12	0.14	0.45
16.17-EDP	16.00	343 > 233	0.9487	11	7	3	16	5	7	0.24	0.79
7.8-EDP	16.24	343 > 141	0.9918	7	6	7	24	20	11	0.30	0.99
13.14-EDP	16.00	343 > 193	0.9934	14	14	12	15	13	22	0.21	0.69
LTB4	8.50	335 > 195	0.9993	4	3	4	17	17	11	0.08	0.27
LTC4	9.05	624 > 272	0.9984	10	3	6	11	5	5	0.20	0.67
PGB2	6.62	333 > 175	0.9999	13	2	1	12	7	9	0.26	0.86
PGJ2	6.62	333 > 175	0.9975	22	3	4	22	11	8	0.75	2.51
PGE2	5.64	351 > 251	0.9997	nd	4	3	nd	4	2	3.88	12.95
PGD2	5.64	351 > 251	0.9979	nd	7	2	2	5	2	4.33	14.45
PGD3	5.14	349 > 313	0.9985	1	1	2	1	1	2	0.10	0.30
PGE3	5.14	349 > 313	0.9985	1	1	2	1	1	2	0.10	0.30
8iso-PGF3α	4.97	351 > 253	0.9960	7	4	3	14	14	16	0.22	0.73
8iso-PGF2α	5.47	353 > 299	0.9936	8	3	2	8	3	2	0.06	0.20
MaR-1	8.09	359 > 177	0.9998	5	2	5	9	4	4	0.21	0.71
PDx	7.67	359 > 153	0.9984	3	1	3	3	3	4	0.05	0.16
TXB3	5.05	367 > 195	0.9981	1	3	2	10	4	2	0.06	0.21
RvD1	6.00	375 > 141	0.9958	1	1	2	1	1	2	0.02	0.07
RvD2	5.56	375 > 141	0.9988	4	2	2	9	7	3	0.12	0.39
12-HETE-d8	15.43	326 > 183		5	6	4	5	6	4	0.002	0.006
ARA	17.34	303 > 259	0.9677	7	6	12	7	11	11	1.96	6.53
EPA	16.92	301 > 257	0.9923	13	15	8	11	16	12	0.28	0.93
DHA	17.27	327 > 283	0.9925	10	13	11	7	10	15	0.18	0.59

**Table 2 molecules-24-02276-t002:** Repeatability of the SPE-LC-MS assay. Global Recovery (SPE-LC-MS). Extraction recovery (SPE) and matrix effect (LC-MS) for spiked plasma and adipose tissue samples (*n* = 3). n.e. (non-evaluated).

Compound	%RSD (*n* = 3)		%Global Recovery		%SPE Recovery		%Matrix Effect	
	Plasma	Adipose	Plasma	Adipose	Plasma	Adipose	Plasma	Adipose
15-HEPE	9	9	91	87	102	88	88	99
18-HEPE	7	6	85	81	107	95	80	85
12-HEPE	10	6	84	94	103	94	82	100
5-HEPE	8	7	84	64	107	90	79	71
11-HEPE	2	6	93	86	115	94	81	92
11-HETE	18	3	75	91	93	96	73	95
5-HETE	15	11	144	155	113	115	199	122
12-HETE	3	6	91	67	97	109	86	58
15-HETE	5	12	86	81	101	108	85	75
20-HETE	9	9	95	92	107	101	89	91
4-HDoHE	7	5	489	442	101	97	494	580
17-HDoHE	2	9	84	78	111	78	76	101
14-HDoHE	9	10	85	66	101	95	84	70
11-HDoHE	8	0	81	81	108	91	75	84
15-OxoETE	7	8	87	54	104	92	84	59
12-OxoETE	3	14	94	52	107	80	90	65
5-OxoETE	15	6	379	294	101	72	344	409
15-HpEPE	2	9	17	22	87	89	10	25
12-HpEPE	0	6	7	18	29	60	25	33
12-HpETE	4	21	27	28	92	90	26	30
5-HpETE	16	11	53	71	64	105	83	67
15-HpETE	0	23	37	44	57	71	66	67
17-HpDoHE	3	12	24	19	86	94	14	21
14.15-DiHETE	7	10	95	90	92	93	103	97
5.15-DiHETE	4	8	92	86	99	94	94	92
5.6-DiHETE	12	8	61	95	75	96	82	99
11.12-DiHETrE	3	12	69	35	82	41	83	83
8.9-DiHETrE	7	18	82	36	92	42	90	84
14.15-DiHETrE	2	7	59	35	91	40	64	85
5.6-DiHETrE	6	10	83	34	93	42	88	80
19.20-DiHDPA	1	15	78	37	91	41	85	87
17.18-DiHETE	1	52	87	59	99	63	88	89
7.17-hydroxiDPA	1	27	74	33	89	42	82	79
11.12-EpETrE	1	19	447	142	107	45	397	341
8.9-EpETrE	3	30	334	121	107	45	309	266
14.15-EpETrE	9	29	179	41	114	38	143	145
5.6-EpETrE	7	20	228	104	103	38	211	276
17.18-EpETE	2	22	86	35	100	50	84	72
11.12-EpETE	3	13	78	37	102	45	76	80
14.15-EpETE	0	16	70	36	100	44	70	81
8.9-EpETE	0	21	69	34	101	47	69	69
19.20-EDP	4	10	185	51	134	45	129	110
10.11-EDP	21	30	394	159	105	46	396	346
16.17-EDP	2	17	469	194	111	53	404	333
7.8-EDP	10	8	238	114	108	50	228	237
13.14-EDP	1	19	471	191	109	50	433	336
LTB4	8	7	82	52	98	96	78	54
LTC4			0	0	0	0	60	65
PGB2	2	6	97	68	115	93	85	73
PGJ2	5	9	81	67	97	88	83	76
PGE2	8	8	81	82	101	90	80	91
PGD2	7	67	128	46	109	94	118	51
PGD3/PGE3	10	n.e.	94	n.e.	94	n.e.	100	n.e.
8iso-PGF3α	14	11	45	33	50	43	103	79
8iso-PGF2α	8	n.e.	86	n.e.	85	n.e.	102	n.e.
MaR-1	3	1	108	73	124	111	87	60
PDx	12	5	97	75	102	103	96	74
TXB3	7	11	48	36	61	55	78	68
RvD1	12	n.e.	85	n.e.	75	n.e.	89	n.e.
RvD2	10	9	63	56	98	86	64	65
12-HETE-d8	15	6	73	60	91	94	76	64
ARA	4	n.e.	100	n.e.	111	n.e.	90	n.e.
EPA	6	n.e.	98	n.e.	106	n.e.	93	n.e.
DHA	3	n.e.	88	n.e.	123	n.e.	71	n.e.

**Table 3 molecules-24-02276-t003:** Hypothesized specific fragments produced from different isomers of ARA. EPA and DHA hydroperoxides based on the chain position of –OOH group.

Parent Mass *m/z*	Family	Isomers	Specific Fragments from α-Hydroxy Cleavage *m/z*	Specific Fragments from Diol *m/z*
359	HpDoHE	4-HpDoHE	101. 99. 71	121. 101. 73
		7-HpDoHE	141. 139. 111	161. 141. 113
		8-HpDoHE	153. 151. 123	173. 153. 125
		10-HpDoHE	181. 179. 151	201. 181. 153
		13-HpDoHE	221. 219. 191	241. 221. 193
		14-HpDoHE	233. 231. 203	253. 233. 205
		16-HpDoHE	261. 259. 231	281. 261. 233
		20-HpDoHE	313. 311. 283	333. 313. 285
333	HpEPE	8-HpEPE	155. 153. 125	175. 155. 127
		9-HpEPE	167. 165. 137	187. 167. 139
		11-HpEPE	195. 193. 165	213. 193. 165
		18-HpEPE	287. 285. 257	307. 287. 259
335	HpHETE	8-HpETE	155. 153. 125	175. 155. 127
		9-HpETE	167. 165. 137	187. 167. 139
		11-HpETE	195. 193. 165	213. 193. 165
		19-HpETE	303. 301. 273	321. 301. 273
		20-HpETE	317. 315. 287	335. 315. 287

**Table 4 molecules-24-02276-t004:** Compounds tentatively identified in adipose tissue and plasma, validation with standards and specific transitions used for our hypothetical identifications.

**Parent Ion *m/z***	**Identified Compounds in Adipose Tissue**		**Identified Compounds in Plasma**		**Specific Transitions *m/z*** (Most Intense-Non Specific)
	**Screening Analysis** (Hypothetic Identification)	**Validation with Standards** (Confirmation/Rejection)	**Screening Analysis** (Hypothetic Identification)	**Validation with Standards** (Confirmation/Rejection)	
**317**	18-HEPE	confirmation	18-HEPE	confirmation	287. 259
	15-HEPE	confirmation	15-HEPE	confirmation	247. 219
	12-HEPE	confirmation	12-HEPE	confirmation	207. 179
	11-HEPE	confirmation	11-HEPE	confirmation	195. 167
	9-HEPE				167. 139
	5-HEPE	confirmation	5-HEPE	confirmation	115. 87
	12-oxoETE	confirmation	12-oxoETE	confirmation	207. 179. (153)
	15-oxoETE	rejection			247. 219. (113)
	5-oxoETE	rejection	5-oxoETE	rejection	115. 87. (203)
319			20-HETE	confirmation	317. 289. (245)
	15-HETE	confirmation	15-HETE	confirmation	247. 219
	12-HETE	confirmation	12-HETE	confirmation	207. 179
	11-HETE	confirmation	11-HETE	confirmation	195. 167
	5-HETE	rejection	5-HETE	rejection	115. 87
	5.6-EpETrE	confirmation	5.6-EpETrE	confirmation	115. 87
	11.12-EpETrE	confirmation			207. 179. 167
	14.15-EpETrE	confirmation	14.15-EpETrE	confirmation	249. 221. 207
343	20-HDoHE		20-HDoHE		313. 285. (187)
	17-HDoHE	confirmation	17-HDoHE	confirmation	273. 245
	16-HDoHE				261. 233
	14-HDoHE	confirmation	14-HDoHE	confirmation	233. 205
	13-HDoHE		13-HDoHE		221. 193
	11-HDoHE	confirmation			193. 165. (149)
			10-HDoHE		181. 153
	8-HDoHE				153. 125. (189). (109)
	4-HDoHE	confirmation	4-HDoHE	confirmation	115. 101. 73
359-341		17HpDoHE		17HpDoHE	297
335	8.15-DiHETE				265. 155
	14.15-DiHETE	confirmation	14.15-DiHETE	confirmation	265. 219. 247. 207
	5.12-DiHETE=LTB4	rejection	5.12-DiHETE=LTB4	rejection	195. 129
	11.18-DiHETE		11.18-DiHETE		289. 165. 193
	5.6-DiHETE	confirmation	5.6-DiHETE	confirmation	115. (145)
	11.12-DiHETE		11.12-DiHETE		234. 222. 193. 165
333-315			12-HpEPE	confirmation	207. 227. 179
351	PGE2-PGD2	confirmation	PGE2-PGD2	confirmation	251
303	ARA	confirmation	ARA	confirmation	259
301	EPA	confirmation	EPA	confirmation	257
327	DHA	confirmation	DHA	confirmation	283
337	14-15DiHETrE	confirmation			207. 129

**Table 5 molecules-24-02276-t005:** MS/MS conditions of the non-targeted LC-MS/MS method developed. More than 100 compounds were classified in 22 groups according to the *m/z* of their parent ions in negative ESI. Compromise collision energies used for each *m/z* are shown. The families of lipid mediators studied in each parent mass of the method are indicated as well as some examples of metabolites.

Parent Mass (*m/z*)	Collision Energy (V)	Family of Lipid Mediators	Some Examples
301.0	25	PUFAs-ω3	EPA
303.0	30	PUFAs-ω6	ARA
317.0	28	HEPEs	5-HEPE. 8-HEPE. 12-HEPE
		EpETEs	17.18-EpETE
		oxoETEs	12-oxoETE. 5-oxoETE
		LTs	LTA4
319.0	30	HETEs	9-HETE. 11-HETE. 15-HETE
		EpETrEs	11.12-EpETrE. 14.15-EpETrE
326.0	27	internal standard	12-HETE-d8
327.0	27	PUFA-ω3	DHA
331.0	30	PGs	PGA3. PGB3
333.0 & 315.0	27	HpEPEs	5-HpEPE. 12-HpEPE
333.0	27	Rvs	RvE2
		PGs	PGJ2. PGB2
335.0 & 317.0	30	HpETEs	11-HpETE. 12-HpETE
335.0	30	DiHETEs	5.12-DiHETE. 17.18-DiHETE
337.0	30	DiHETrEs	11.12-DiHETrE. 8.9-DiHETrE
343.0	28	EDPs	10.11-EDP
		HDoHEs	4-HDoHE. 17-HDoHE. 20-HDoHE
349.0	30	PGs. k-PGs	PGD3. PGE3. 15k-PGE2
		Rvs	RvE1
		LXs	LXA5
351.0	28	PGs. k-PGs	PGE2. PGF3α. 15k-PGF2α
		TXs	TXA2
		LXs	LXA4. LXB4
353.0	30	PGs. k-PGs	PGF2α. 15k-PGF1α
359.0 & 341.0	26	HpDoHEs	4-HpDoHE. 17-HpDoHE
359.0	26	PDs	PDx
		MaRs	7-MaR1
361.0	28	DiHDPAs	19.20-DiHDPA
367.0	20	PGs. k-PGs	PGG2. 6k-PGE1
		TXs	TXB3
369.0	30	TXs	TXB2
		k-PGs	6k-PGF1α
375.0	27	Rvs	RvD1. RvD2
624.0	26	LTs	LTC4

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
