# Peer review of "Non-Targeted LC-MS/MS Assay for Screening Over 100 Lipid Mediators from ARA, EPA, and DHA in Biological Samples Based on Mass Spectral Fragmentations"

_molecules, 2019, doi:10.3390/molecules24122276_

Round 1

Reviewer 1 Report

The manuscript by Dasilva et al, titled "Non-targeted LC-MS/MS assay for screening over 100 lipid mediators from ARA, EPA, and DHA in biological samples based on mass spectra fragmentations" focuses on the development of an untargeted LC-MS method to detect lipid mediators. While the authors provide a number of experimental and proposed fragmentation spectra "library" which might be of particular help to identify novel lipid mediators, the manuscript suffers from shortcomings. I strongly recommend that the authors thoroughly improve clarity, spelling and grammar, and emphasize better the novelty of their work before resubmitting.

Major points:

Spelling and grammar must be improved. Examples where the conveyed message is unclear are, "due to they are/have", l.47/448/452; "The production these", l.51; "The most studied derived from ARA are", l.56; "freeze-tissue were cut", l.183; "or the enhance of", l.268; "stechiometric", l.325; "we could not hypothesize the presence", l.520f; "and 6 ones", l.525; "rejec", tab5; "avoiding the information loss due to the use of predefined ions list", l.545f.

Nomenclature is inconsistent. location of double bonds/functional groups should consistently be followed by a slash (not as in table 2)

Methods are not clearly described: What is "total ion scan" mode? Is it Fullscan, or some sort of PRM (since fragment spectra were acquired)? There is no "drying gas" with Thermo ion sources. Were the spectra acquired in the ion trap or the orbitrap (resolution?). What isolation was used for MS2?

Data analysis is not described at all. How was the data analyzed? The concentrations of the measured lipid mediators in plasma and in adipose tissue should be reported in the supplement for comparison.

Minor points:

LipidMAPS is frequently referenced, in regards to downloaded fragment spectra. The authors should provide the link to the spectral library. Additionally, for fragment spectra, information about spray voltage (-4500V) is irrelevant and can be omitted. "LIPID MAPS" should be written using correctly

The instrument used is not a "linear ion trap" (l. 202)

inconsistent citation style (cit 28-29-30)

Table 1: parent mass should have at least one decimal

Tables 1/2/3: if table stretches across multiple pages, it would be helpful to have the headings on every page

Spectra are frequently described as "product ion spectra at 319 m/z", when it should be "product ion spectrum of m/z 319"

Fig 6, fragmentation pathway overlays the compound names

Fig 9, in contrast to all other spectra, the fragmentation pathway is missing.

Author Response

June 6th, 2019

Dear Sir/Madame:

The manuscript entitled “Non-targeted LC-MS/MS assay for screening over 100 lipid mediators from ARA, EPA, and DHA in biological samples based on mass spectra fragmentations(Molecules, special issue of Mass Spectrometry Based Lipidomics. ) has been reviewed and corrected according to the editor and reviewers comments. We are very grateful for the comments and suggestions that have contributed to improve the manuscript.

Response to Reviewer 1 comments:

Point 1: Extensive editing of English language and style are required.

Response 1: An extensive revision of language, spelling, grammar and style has been done.

Point 2: While the authors provide a number of experimental and proposed fragmentation spectra "library" which might be of particular help to identify novel lipid mediators, the manuscript suffers from shortcomings. I strongly recommend that the authors thoroughly improve clarity, spelling and grammar, and emphasize better the novelty of their work before resubmitting.

Response 2: As mentioned above, an extensive revision of language, spelling, grammar and style has been done.

As regards to novelty, we have emphasized it in the abstract and conclusions. We suggested new specific transitions to identify lipid mediators, especially for those derived from EPA and DHA which are difficult to identify due to the high number of isomers usually present in samples. And, we also achieved a high score of success in identifications by using a LIT without the high mass resolution of an Orbitrap. The method allowed the identification of these metabolites without losing information by the use of predefined ions list.

Point 3: Spelling and grammar must be improved. Examples where the conveyed message is unclear are, "due to they are/have", l.47/448/452; "The production these", l.51; "The most studied derived from ARA are", l.56; "freeze-tissue were cut", l.183; "or the enhance of", l.268; "stechiometric", l.325; "we could not hypothesize the presence", l.520f; "and 6 ones", l.525; "rejec", tab5; "avoiding the information loss due to the use of predefined ions list", l.545f.

Response 3: We have performed the following changes along the manuscript: “due to they are/have” into “because they are/have”; "The production these" into “The production of these”; “The most studied derived from ARA are” into “The most studied compounds derived from ARA are”; “freeze-tissue were cut” into “150 mg of frozen tissues were cut”; “or the enhance of” into “Among the main causes stand out: the ion suppression, and the overproduction of”; “stechiometric” into “stoichiometric”; “we could not hypothesize the presence” into “we could not issue the hypothesis of the presence”; “and 6 ones” into “and 6 isomers”; “rejec” into “rejection”; and “avoiding the information loss due to the use of predefined ions list” into “without losing information by the use of predefined ions list”.

Point 4: Nomenclature is inconsistent. Location of double bonds/functional groups should consistently be followed by a slash (not as in table 2)

Response 4: According to the reviewer suggestion we modified along the manuscript the nomenclature of lipid mediators in order to be consistent. Following IUPAC rules, numbers indicating position of functional groups were separated by a dash from the name of the compound, and numbers were separated by commas. For instance, in the previous version we had written 17HDoHE or 17-18DiHETE. In the revised version we have changed to: 17-HDoHE and 17,18-DiHETE. Then, we’ve modified the text, and also Tables and Fig 1a and Fig 1b.

Point 5: Methods are not clearly described: What is "total ion scan" mode? Is it Fullscan, or some sort of PRM (since fragment spectra were acquired)? There is no "drying gas" with Thermo ion sources. Were the spectra acquired in the ion trap or the orbitrap (resolution?). What isolation was used for MS2?

Response 5: We have now explained in the text the meaning of Total Ion Scan Mode: Total Ion Scan Mode is a full scan detection mode done over a group of parent masses previously isolated. Nitrogen as a drying gas was corrected to nitrogen as a nebulizing gas. Spectra were acquired in the ion trap, not Orbitrap.

Point 6: Data analysis is not described at all. How was the data analyzed? The concentrations of the measured lipid mediators in plasma and in adipose tissue should be reported in the supplement for comparison.

Response 6: We’ve now included a new section, “2.7 Data analysis” in the Experimental section. Here we’ve described the use of R free software (version 3.2.4) for calculating means, standard deviations, least-squares linear regression curves, and percentages used to validate methodology were performed.

Regarding concentrations of lipid mediators, the scope of this work was mainly qualitative to identify as many compounds as possible with the proposed method. So, we did not calculate their quantitative values.

Point 7: LipidMAPS is frequently referenced, in regards to downloaded fragment spectra. The authors should provide the link to the spectral library. Additionally, for fragment spectra, information about spray voltage (-4500V) is irrelevant and can be omitted. "LIPID MAPS" should be written using correctly

Response 7: We corrected spelling from “lipidmaps” to “LIPID MAPS in the manuscript and Supp. Material captions, we have provided the link to the library the first time we reference LIPID MAPS in the text, and we’ve deleted information of spray voltage in the text and Supp. Material.

Point 8: The instrument used is not a "linear ion trap" (l. 202)

Response 8: The mass spectrometer we used is the LTQ Velos Pro (Thermo Fisher). So, we have modified the information in the text for explaining that it is a “dual-pressure linear ion trap”

Point 9: inconsistent citation style (cit 28-29-30)

Response 9: Cites were modified according to the journal style:

(28)     Cavalca, V.; Minardi, F.; Scurati, S.; Guidugli, F.; Squellerio, I.; Veglia, F.; Dainese, L.; Guarino, A.; Tremoli, E.; Caruso, D. Anal. Biochem. 2010, 397 (2), 168–174.

(29)     Kortz, L.; Dorow, J.; Becker, S.; Thiery, J.; Ceglarek, U. J. Chrom. B. 2013,  927, 209-213

(30)     Strassburg, K.; Huijbrechts, A. M. L.; Kortekaas, K. A.; Lindeman, J. H.; Pedersen, T. L.; Dane, A.; Berger, R.; Brenkman, A.; Hankemeier, T.; van Duynhoven, J.; Kalkhoven, E.; Newman, J. W.; Vreeken, R. J. Anal. Bioanal. Chem. 2012, 404 (5), 1413–1426.

Point 10: Table 1: parent mass should have at least one decimal

Response 10: Table 1 has been corrected and values include one decimal in the actual version.

Point 11: Tables 1/2/3: if table stretches across multiple pages, it would be helpful to have the headings on every page

Response 11: We agree with the reviewer. As Tables are done in an editable format, it would be better that editors of the journal decide how to edit Tables when the final version of the manuscript is done.

Point 12: Spectra are frequently described as "product ion spectra at 319 m/z", when it should be "product ion spectrum of m/z 319"

Response 12: We’ve corrected these terms in each spectra and m/z was placed before the number along the manuscript and Supp. Material captions.

Point 13: Fig 6, fragmentation pathway overlays the compound names

Response 13: Figure 6 has been modified and the name is not overlayed in the actual version.

Point 14: Fig 9, in contrast to all other spectra, the fragmentation pathway is missing.

Response 14: We double-checked and Fig. 9 shows the spectrum and fragmentation pathway in the actual version.

Trusting that you will find this manuscript amenable to your requirements on its present form, and waiting for your reply, I remain.

Kind regards,

Gabriel Dasilva Alonso

Reviewer 2 Report

This is a very interesting paper about oxylipi analysis with a mixture of classical targeted approach with non targeted one. The hypothesis done for general fragmentation rule are interesting and useful. It is well written and quite easy to follow.

Some details are needed (see after)

Material & method :

2.3 : please prefer “sample preparation” instead of “SPE” for the title

Some details are needed:

- Add IST quantities

-Precise the volume for the MeOH water (30 : 70)  :added to plasma before SPE , it is not clear

- did you do a centrifugation before the SPE (it is usually necessary)

2.4 :

I didn’t completely understand how did you optimize the collision energy and what was the “other works” you spoke about? please explain

What are the “example “showed in the last column in table 1 ? why did you choose these one?

2.5 :

Did you measure LOD and LOQ with extracted calibration curves ?

On the top of S/N did you check the accuracy to determine LOD and LOQ ?

How did you manage to measure the yield of recovery of molecule which are present in plasma of adipose tissue ?

Result & Discussion :

3.1 :

Please compare the data you abtained for LOD and LOQ with literature

About the quantification of ARA, EPA and DHA are you sure to recover all the fatty acid with this chromatographic system? Usually they stick on a C18 column, please show a blank chromatogram

3.2 :

Please indicate the link for the lipid map web site

Please discussed and compare your data with Massoodi et al 24 publication who used a similar approach.

Author Response

June 6th, 2019

Dear Sir/Madame:

The manuscript entitled “Non-targeted LC-MS/MS assay for screening over 100 lipid mediators from ARA, EPA, and DHA in biological samples based on mass spectra fragmentations(Molecules, special issue of Mass Spectrometry Based Lipidomics. ) has been reviewed and corrected according to the editor and reviewers comments. We are very grateful for the comments and suggestions that have contributed to improve the manuscript.

Response to Reviewer 2 comments:

Point 1: Moderate English changes required

Response 1: An extensive revision of language, spelling, grammar and style has been done.

Point 2: Material & method: 2.3: Please prefer “sample preparation” instead of “SPE” for the title. Add IST quantities. Precise the volume for the MeOH water (30 : 70): added to plasma before SPE, it is not clear. Did you do a centrifugation before the SPE (it is usually necessary)

Response 2:

2.3 Title has been modified from SPE into Sample preparation by Solid Phase Extraction. The quantity of IST was included in the text (500 ng/mL) and also the volume of MeOH:water (1300 µL).

Finally, before SPE a centrifugation was done and remaining solution was loaded into conditioned Oasis-HLB cartridges.

Point 3: Material & method: 2.4: I didn’t completely understand how did you optimize the collision energy and what was the “other works” you spoke about? please explain. What are the “example “showed in the last column in table 1 ? why did you choose these one?

Response 3:

We had used the words “other works” to refer to previous papers. We’ve modified that in the text. To set up collision energies in our method, we considered the experimental data that we obtained for 65 analytical standards, and the most common collision energies found in literature for other compounds with the same m/z than our 65 standards. Then, we did not set up the optimal collision energy for each standard but the best one for most of compounds with the same m/z. Our goal was to create a methodology to do a screening of samples and identify as many compounds as possible.

The examples shown in last column of Tab. 1 are compounds with the m/z of the parent mass selected for the method. These are the most representative from each family of lipid mediators.

Point 4: Material & method: 2.5: Did you measure LOD and LOQ with extracted calibration curves ? On the top of S/N did you check the accuracy to determine LOD and LOQ ? How did you manage to measure the yield of recovery of molecule which are present in plasma of adipose tissue ?

Response 4:

Calibration curves were done to validate the linear range of response for each analyte, not for LOD and LOQ. To determine these limits, we used the standard at the lowest concentration level for each compound. Firstly, we measured the signal of peak and the signal of the noise around the peak. Secondly, we calculated the ratio S/N. We performed these measures by quadruplicate to check also the accuracy of these values.

For recovery experiments, plasma samples or adipose samples were spiked with a set of 65 analytical standards. Other set of samples were not spiked. Then, we compared the signal for every compound in the spiked sample with the signal obtained when we directly analyze the standard. If the compound was naturally present in the sample, we have subtracted the signal of the spiked sample with the non-spiked one in order to not overestimate the recovery.

Point 5: Result & Discussion: 3.1: Please compare the data you abtained for LOD and LOQ with literatura. About the quantification of ARA, EPA and DHA are you sure to recover all the fatty acid with this chromatographic system? Usually they stick on a C18 column, please show a blank chromatogram

Response 5:

We compared our LOD and LOQ with literature and we included references in the text (ref 16-28-35).

Regarding the extraction of PUFAs (ARA, EPA and DHA), the percentage of MeOH:water (30:70) that we used in the LLE before the SPE was crucial to achieve good recoveries. In a previous work (1) we checked the influence of the MeOH percentage and we observed higher recoveries for PUFAs when we used 50:50; however, we obtained worst recoveries for other lipid mediators that are more polar than PUFAs. With percentages of 15:85 or less, we achieved bad recovery results for PUFAs and slightly increased recoveries for more polar lipid mediators. Then, we decided to continue with 30:70 because it was a good solution for the wide range of polarities.

1.        Dasilva G, Pazos M, Gallardo JM, Rodríguez I, Cela R, Medina I. Anal Bioanal Chem. 2014;406.

Point 6: Result & Discussion: 3.2: Please indicate the link for the lipid map web site. Please discussed and compare your data with Massoodi et al 24 publication who used a similar approach.

Response 6:

We provided the link to LIPID MAPS: https://www.lipidmaps.org in section 3.2

We discussed our data with Masoodi et al in the Introduction section. We developed a similar approach for the screening of lipid mediators. And, to the best of our knowledge, there are no other works that analyzed these compounds based on screening MS/MS methods. Masoodi used an Orbitrap to obtain high resolution in parent masses for identification and LIT to validate results in MRM. We suggested new specific transitions to identify lipid mediators, especially for those derived from EPA and DHA, and we achieved a high score of success in identifications by using a LIT without the high mass resolution of an Orbitrap. Further improvements of these techniques are needed in order to increase the number of target metabolites that can be accurately identified in a single screening step before the quantitative analysis.

Trusting that you will find this manuscript amenable to your requirements on its present form, and waiting for your reply, I remain.

Kind regards,

Gabriel Dasilva Alonso

Round 2

Reviewer 1 Report

The authors have answered most of my concerns from their previous version and the manuscript has improved substantially.

There still remain some questions about "total ion scan", a term that I have never come across before, despite being familiar with Thermo LTQ instruments.

In l.119 authors describe total ion scan as "a full scan (...)over a group of parent masses previously isolated", which would be similar to a multiplexed t-SIM, but this scan mode is not possible in an ion trap (as precursor selection is the same procedure as m/z detection). Then again, authors did product ion scans, so was the detection constant product ion scanning, such as PRM? In methods, authors describe detection as "full scan (..) parent masses in a single segment", which sounds like DDA mode. Table 2 states "MSMS quantification", which sounds like MRM? Please clarify the scan modes used.

In Table 1, the first decimal is 0 for all compounds, which is redundant, and also incorrect (e.g., EPA has accurate m/z of 301.2173). Depending on the MS2 isolation window (please state in methods), this could lead to significant loss of signal during precursor selection.

Data analysis does not describe identification of the analytes, in my experience, even MRM analysis of lipid mediators gives multiple peaks and interferences for many compounds, please state how the correct peaks were identified, and how chromatographic peaks were extracted (from MS1, or MS2?)

How exactly was recovery determined? This is typically only a concern with quantitative methods, furthermore how does spiking with analytical standards (I assume these standards are also present in the sample) give a measure of recovery? Recovery is typically determined by spiking internal standards before and after sample preparation.

Author Response

June 13rd, 2019

Dear Sir/Madame:

The manuscript entitled “Non-targeted LC-MS/MS assay for screening over 100 lipid mediators from ARA, EPA, and DHA in biological samples based on mass spectra fragmentations(Molecules, special issue of Mass Spectrometry Based Lipidomics. ) has been reviewed and corrected according to the reviewer comments. We are very grateful for the comments and suggestions that have contributed to improve the manuscript.

Response to Reviewer 1 comments:

Point 1: There still remain some questions about "total ion scan", a term that I have never come across before, despite being familiar with Thermo LTQ instruments

In l.119 authors describe total ion scan as "a full scan (...) over a group of parent masses previously isolated", which would be similar to a multiplexed t-SIM, but this scan mode is not possible in an ion trap (as precursor selection is the same procedure as m/z detection). Then again, authors did product ion scans, so was the detection constant product ion scanning, such as PRM? In methods, authors describe detection as "full scan (..) parent masses in a single segment", which sounds like DDA mode. Table 2 states "MSMS quantification", which sounds like MRM? Please clarify the scan modes used.

Response 1:

We have probably named incorrectly the scan mode of the used method. Then, we changed “Total Ion Scan Mode” to “full scan mode” and we named the method as a “non-targeted LC-MS/MS method” along the manuscript.

As we explained in 2.4. LC-MS/MS section, MS method followed the following steps: (1st) 22 parent masses were set in the method, (2nd) each parent mass was fragmented according to the collision energy indicated, and (3rd) acquisition of fragments was done in full scan mode ranging from 90 to 400 m/z

Point 2: In Table 1, the first decimal is 0 for all compounds, which is redundant, and also incorrect (e.g., EPA has accurate m/z of 301.2173). Depending on the MS2 isolation window (please state in methods), this could lead to significant loss of signal during precursor selection.

Response 2:

We agree with the referee that would be desirable to include as much decimals as possible in parent masses. However, our instrument has lower accuracy than TOF, Orbitrap and probably other equipment’s. In our method all parent masses have two decimals which are zero, and that is why we had not included decimals in Tab 1 in the first version. Regarding the isolation width for masses, it was set 1 for all of them. This parameter was now included in the manuscript in 2.4 LC-MS/MS section.

Point 3: Data analysis does not describe identification of the analytes, in my experience, even MRM analysis of lipid mediators gives multiple peaks and interferences for many compounds, please state how the correct peaks were identified, and how chromatographic peaks were extracted (from MS1, or MS2?)

Response 3:

To clarify this point we’ve rewritten the section 2.6: Qualitative screening of samples and validation in order to better explain data analysis. Plasma and adipose tissue were subjected to SPE-LC-MS/MS. Chromatographic peaks for each parent mass were extracted from MS2 selecting quantitative and selective transitions for each lipid mediator according to the fragmentation pattern previously studied. The spectra of each chromatographic peak was studied and compared with the fragmentation pattern to tentatively identify compounds in samples. Then, other samples were spiked before extraction with a pool of 65 standards described in the chemical section. Spiked samples were analyzed under the same LC-MS/MS conditions than non-spiked ones, and hypothetical identifications were validated using the information of retention times and fragmentation in standards and samples.

We agree with the referee that proper identification of lipid mediators is difficult and that is one of the goals of the work. The presence of many isomers, sometimes co-eluting isomers give rise to different peaks that may lead to false positive identifications. Then, we did not trust only retention times and quantification transitions. We studied the whole spectra for identification and then we suggested quantification transitions (Tab. 2) for each compound. Transitions were suggested in terms of selectivity for the compound and also sensitivity of the signal.

Point 4: How exactly was recovery determined? This is typically only a concern with quantitative methods, furthermore how does spiking with analytical standards (I assume these standards are also present in the sample) give a measure of recovery? Recovery is typically determined by spiking internal standards before and after sample preparation.

Response 4:

To explain better how recovery was determined we’ve rewritten section 2.5: Different recovery experiments were performed to determine global recovery, SPE recovery and matrix effect. (a) The percentage of global recovery was evaluated comparing the signal for each metabolite between: spiked plasma and adipose tissue samples before preparation, and a pool of analytical standards at the same level of concentration. . (b) The percentage of SPE recovery was determined comparing samples spiked before preparation with samples spiked after the SPE step. (c) Matrix effect was evaluated comparing samples spiked after the SPE extraction with the pool of analytical standards. In any case, when metabolites were already present in blank samples (non-spiked samples), the area of chromatographic peaks in spiked samples was subtracted with the area in blanks. Experiments were done in triplicate and the repeatability of the entire method was expressed as percentage of relative standard deviation (%RSD).

Trusting that you will find this manuscript amenable to your requirements on its present form, and waiting for your reply, I remain.

Kind regards,

Gabriel Dasilva Alonso